

# Facilitating the structural characterisation of non-canonical amino acids in biomolecular NMR

Sarah Kuschert[1], Martin Stroet[2], Yanni Ka-Yan Chin[1], Anne Clair Conibear[3], Xinying Jia[1], Thomas Lee[2], Christian Reinhard Otto Bartling[4], Kristian Strømgaard[4], Peter Güntert[5], Karl Johan Rosengren[6], Alan Edward Mark[2] and Mehdi Mobli[1]

[1]Centre for Advanced Imaging, The University of Queensland, Brisbane, QLD 4072, Australia

[2]School of Chemistry & Molecular Biosciences, The University of Queensland, Brisbane, QLD 4072, Australia

[3]Institute of Applied Synthetic Chemistry, Technische Universität Wien, Getreidemarkt 9/163, Wien 1060, Vienna, Austria

[4]Department of Drug Design and Pharmacology, University of Copenhagen, Universitetsparken 2, 2100 Copenhagen, Denmark

[5]Laboratory of Physical Chemistry, ETH Zürich, 8093 Zürich, Switzerland; Institute of Biophysical Chemistry, Center for Biomolecular Magnetic Resonance, Goethe University Frankfurt, 60438 Frankfurt am Main, Germany; Department of Chemistry, Tokyo Metropolitan University, Hachioji, Tokyo 192-0397, Japan

[6]School of Biomedical Sciences, The University of Queensland, Brisbane, QLD 4072, Australia

*Correspondence to*: Mehdi Mobli (m.mobli@uq.edu.au)



**Abstract**
Peptides and proteins containing non-canonical amino acids (ncAAs) are a large and important class of biopolymers. They
include non-ribosomally synthesised peptides, post-translationally modified proteins, expressed or synthesised proteins
containing unnatural amino acids, and peptides and proteins that are chemically modified. Here, we describe a general
procedure for generating atomic descriptions required to incorporate ncAAs within popular NMR structure determination
software such as CYANA, CNS, Xplor-NIH and ARIA. This procedure is made publicly available via the existing Automated
Topology Builder (ATB) server (atb.uq.edu.au) with all submitted ncAAs stored in a dedicated database. The described
procedure also includes a general method for linking of sidechains of amino acids from CYANA templates. To ensure
compatibility with other systems, atom names comply with IUPAC guidelines. In addition to describing the workflow, 3D
models of complex natural products generated by CYANA are presented, including vancomycin. In order to demonstrate the
manner in which the templates for ncAAs generated by the ATB can be used in practice we use a combination of CYANA and
CNS to solve the structure of a synthetic peptide designed to disrupt Alzheimer-related protein-protein interactions.
Automating the generation of structural templates for ncAAs will extend the utility of NMR spectroscopy to studies of more
complex biomolecules, with applications in the rapidly growing fields of synthetic and chemical biology. The procedures we
outline can also be used to standardise the creation of structural templates for any amino acid and thus have the potential to
impact structural biology more generally.

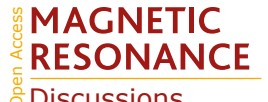

## 1 Introduction

The 20 genetically encoded amino acids, together with selenocysteine and pyrrolysine, provide the basis for most proteins and peptides that make up the machinery of life (Liu and Schultz, 2010; Huang et al., 2010; Bullwinkle et al., 2014). The use of just 22 amino acids, however, limits the structural complexity and functional diversity that can be achieved. The chemical and functional diversity of ribosomally synthesised proteins is further expanded by posttranslational modification (PTM), including processes such as acylation, methylation, phosphorylation, oxidation and epimerisation (Aebersold et al., 2018; Barber and Rinehart, 2018; Walsh et al., 2005). Non-ribosomal synthesis pathways also expand chemical diversity, leading to the production of typically short peptides containing non-canonical amino acids (ncAAs), as well as backbone or sidechain cyclisation (Link et al., 2003; Tharp et al., 2020; Caboche et al., 2008; Goodrich and Frueh, 2015; Martínez-Núñez and López, 2016; Strieker et al., 2010). Such non-ribosomal peptides (NRPs) are prevalent in bacteria and fungi, which produce a wide range of bioactive peptides (Caboche et al., 2008; Marahiel, 2009). In addition to pathways found in nature, chemical synthesis, enzymatic modification, genetic code expansion and site-selective biorthogonal transformations are increasingly used to introduce novel ncAAs and PTMs into peptides and proteins (Bondalapati et al., 2016; Chuh et al., 2016; Conibear, 2020; Hoyt et al., 2019; Thompson and Muir, 2020) for both investigations of peptide structure and function as well as in the design and optimisation of pharmaceuticals (Noren et al., 1989; Coin, 2018; Hoesl and Budisa, 2011; Johnson et al., 2010; Wang et al., 2001; Wang et al., 2020; Zou et al., 2018).

Despite the prevalence of ncAAs and their importance in determining the functional properties of both naturally occurring and synthetic peptides, the structural characterisation of peptides and proteins containing ncAAs remains challenging. For example, of the almost 200 000 structures in the Protein Data Bank (PDB), only 11 677 were annotated with a PTM (Craveur et al., 2014). This paucity of protein structures bearing ncAAs also results in them being excluded from machine learning and structure prediction algorithms, as there is an insufficient training set. Techniques used to determine the structure of proteins and peptides (X-ray diffraction, cryo-electron microscopy and nuclear magnetic resonance (NMR)) rely heavily on modelling software to transform the experimental data into structural models (Mal et al., 2002). The amount of data that can be collected experimentally in NMR is, in general, insufficient to determine the structure of a given peptide or protein directly. Instead, a representation of the spatial arrangement of the atoms within each amino acid, together with a description of the interactions between sets of atoms, is required to translate a set of experimental restraints into a three-dimensional (3D) structure (Mal et al., 2002). Most molecular modelling packages contain only the 20 canonical amino acids and a modest selection of the most common ncAAs. This is true for both general molecular dynamics simulation packages (such as AMBER, CHARMM, GROMACS and GROMOS) as well as software dedicated to structure refinement such as Xplor-NIH (Bermejo and Schwieters, 2018), CNS (Brunger et al., 1998), ARIA (Mareuil et al., 2015; Allain et al., 2020) or CYANA (Guntert and Buchner, 2015; Guntert et al., 1997). Despite the various input formats required for the different structural calculation software, the internal representation of interatomic interactions is in principle the same, or closely related.

Peptides containing ncAAs are ideally suited to NMR structural characterisation as they are small in size, and often contain sidechain links that induce local structure (Hamada et al., 2010; Weber et al., 1991). Nevertheless, their structural characterisation by NMR is often restricted to measurements of a limited set of NOEs, hydrogen bonds or backbone chemical shifts to support specific geometries (Mendive-Tapia et al., 2015; Umstatter et al., 2020). Alternatively, NMR analysis is omitted altogether in favour of lower resolution methods such as CD spectroscopy (De Araujo et al., 2022; Wu et al., 2017). The dearth of high-resolution NMR structures in this class of molecules likely stems, at least in part, from the difficulty of obtaining high-quality atomic representations of the ncAAs required for computer assisted structure determination.



Recently the handling of ncAAs and small molecules in CYANA was addressed by Yilmaz et al. (Yilmaz and Guntert, 2015),
who developed CYLIB, an algorithm that enables automated template generation for ncAAs and small molecules, provided
that a suitable input geometry is available (CIF or Mol2 file). Here, the quality of the input geometry is critical, as the algorithm
does not perform any optimisation of the structure. CYLIB has internal procedures for creating the appropriate branch structure
and ring-closures required by CYANA, however, practical aspects of working with ncAA-containing peptides and proteins
such as consistent atom naming and sidechain linkages are not addressed by CYLIB.
For the algorithms operating in Cartesian space, topology builders have been created which take simplified representations of
amino acids and other molecules as input and infer topological information based on a set of internal rules (Schmid et al.,
2012; Van Der Spoel et al., 2005; Wang et al., 2006). The challenge for incorporating ncAAs lies in the use of specific atom
names to infer bonded and/or non-bonded interactions, as well as assumptions regarding how individual amino acids in a
peptide chain are linked or terminated. While in principle almost any molecule can be represented, users are often unaware of
the assumptions that have been embedded in the codes, or how to incorporate new atom types into the associated files which
contain the parameter definitions needed by the builders to interpret them. Furthermore, common approaches to reduce
complexity such as using atom names to infer interaction type, work well when dealing with a small subset of chemical space
(such as the 20 canonical amino acids). However, linking atom names to specific interactions rapidly leads to a combinatorial
problem, and an explosion in terms, as new classes of interactions are introduced. The addition of a single new atom type can
often require the definition of hundreds of interactions to ensure compatibility with the existing framework.
In attempting to simplify the input for standard (common) cases, the authors of many programs have made the treatment of
non-standard cases progressively more challenging. We have set out to address this problem and provide a generic mechanism
to generate complete topological information for amino acids and related molecules of interest in a robust and reproducible
manner. This can be readily translated into inputs for various simulation and structure refinement packages. Our approach
leverages the capability of the Automated Topology Builder (ATB), a publicly accessible webserver that provides optimised
geometries, validated force field parameters and topology files for a range or popular molecular dynamics (MD) simulation
and structure refinement packages (Stroet et al., 2018; Koziara et al., 2014; Malde et al., 2011). The ATB is provided free to
academic users who can both download existing molecules and submit new structures. In the case of a new structure the user
can submit a set of Cartesian coordinates in, Protein Data Bank (PDB) format together with the net charge on the molecule.
Within the ATB, the geometry of the molecule is first optimised using quantum mechanical (QM) calculations. A topology is
then generated by combining the results with a set of empirical rules (Malde et al., 2011). A variety of formats can be used as
inputs, the most basic being an initial set of Cartesian coordinates in Protein Data Bank (PDB) format and the net charge on
the molecule.
Here, we describe an extension of the ATB that allows users to directly generate template files for a number of popular NMR
structure determination software packages. A template recognition approach has been developed that recognises ncAAs (rather
than "ligands") as forming part of a peptide chain. This allows "building block" files compatible with different software to be
generated, including 3D geometries and templates with atom names that adhere to IUPAC standards. We further introduce a
general method for linking sidechains of amino acids in CYANA and CNS. The utility of the procedure is demonstrated by
generating models of a number of natural products containing complex ncAAs, as well as solving the structure of a sidechain
cyclised synthetic peptide. We further use this method to create a publicly accessible and expanding repository of amino acids
within the ATB. The database currently holds templates for the 20 genetically encoded amino acids with different termini,
common post-translationally modified amino acids, as well as the entire content of the SwissSidechain database (230 entries
in both D- and L- form of amino acids) which includes a wide range of ncAAs (Gfeller et al., 2013).

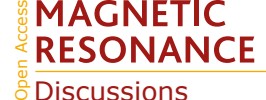

**2 Methods**

**2.1 Outline**

The overall workflow starts with the generation of 3D coordinates of a ncAA in a suitable amino acid template and format (described below) for submission to the ATB, where geometries are optimised and a number of coordinate, parameter, topology and template files are generated. The CYLIB algorithm is used internally by the ATB to generate the CYANA template files from an intermediate Chemical Component Dictionary (CCD) Crystallographic Information Files (CIF) file. All atom names are updated to adhere to IUPAC standards. The resulting outputs are made available to the user in CYANA or CNS format and stored on the ATB server (Fig. 1). Optionally, where sidechain links are required for CYANA templates, these can be generated using the CYANA Lib Linker (atb.uq.edu.au/cyana_linker). Each component of the workflow is described in detail below.

While many pre-calculated ncAAs have been generated as described here and are stored within the ATB repository, the protocol outlined below can be followed by a user to initiate the generation of parameters (templates, topologies, etc.) for any new ncAA. New submissions of ncAAs to the ATB will be added to the existing database and thus the repository will become progressively more complete.

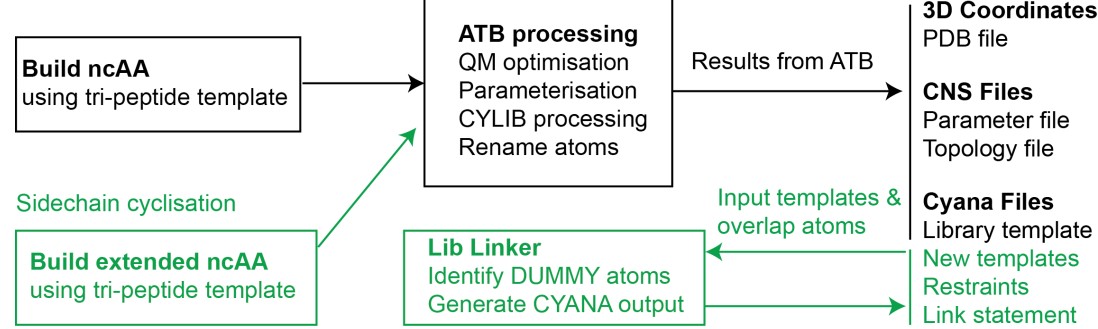

**Figure 1.** Workflow for generating files for NMR structural calculation of peptides and proteins that contain ncAAs. The input file must be generated using the described template format (Fig. 2) and used as input to the ATB webserver, which then recognises the input as an amino acid and excises the necessary portion to generate the required input files for CYANA and CNS-based structure determination softwares (shown in black on the right). To produce templates suitable for sidechain linkage in CYANA (shown in green), an extended ncAA template containing atoms that extend beyond the linking bond is built (Fig. 4). Two such templates are used as input to the CYANA Lib Linker interface of the ATB, where the user also defines which atoms from the first template are present in the second template and vice versa. The "overlap" atoms of the extension are changed to DUMMY atoms to produce a new CYANA template file (.lib file) and short upper distance restraints are generated between sets of overlap atoms (.upl file). A link statement (to be added to the sequence file (.seq) in CYANA) is also produced to remove the repulsion between the two atoms that are to be linked.



### 2.2 Defining the group or amino acid of interest and submission to the ATB

A core feature of the protocol is how the specific group or amino acid of interest is defined and automatically recognised. To
ensure an appropriate chemical environment for the parameterisation of the group of interest within a peptide chain, a series
of structural templates have been defined (Fig. 2). The general form of the peptide template is Ace-Ala-X-Ala-NMe, where
Ace is an N-terminal acetyl capping group, NMe is a C-terminal N-methyl amide capping group, and the central portion of the
structure, denoted X, is the chemical group or amino acid of interest (Fig. 2a). Similarly, structures of the form X-Ala-NMe or
Ace-Ala-X are recognised as representing an N-terminal or C-terminal residue respectively (Fig. 2b and 2c). This template
format allows the molecule to be identified as an amino acid, and the portion 'X' to be excised when generating the parameter
files. Each processed entry is associated with a unique molecule ID (MOLID) within the ATB database.
Submissions to the ATB require an unambiguous molecular representation that includes 3D coordinates (PDB format with all
hydrogen atoms present) along with the net charge. There is no requirement that the input geometry is optimised and thus may
contain clashes, non-ideal bond lengths etc. The stereochemistry, atom information and bonded connectivities, however, must
be specified correctly in the input file. The ATB submission page includes tools that will generate 3D coordinates from an
embedded 2D drawer (JSME (Bienfait and Ertl, 2013)) or a SMILES input. Existing entries within the ATB database can also
be loaded directly into the 2D drawing tool and modified. Documentation and ATB MOLIDs for the structural templates can
be found at github.com/ATB-UQ/CYANA-Examples. There is, in principle, no limit to the size of the group of interest.
However, for computational reasons, the default limit for density functional theory (DFT) calculations to be performed as part
of the parameterisation by the ATB is 50 atoms and the limit for semi-empirical QM processing is 500 atoms. Note that the
most significant difference between DFT and semi-empirical QM processing by the ATB is the atomic charge model, which
is not relevant for CYANA calculations as electrostatic interactions are not considered. While the group of interest is not
required to be a canonical amino acid, it must be able to be incorporated into a peptide chain via peptide bonds. The procedures
used by the ATB to parameterise molecules and the steps taken to validate these parameters have been described in detail
elsewhere (Malde et al., 2011; Stroet et al., 2018).
All amino acid entries are listed as such on the ATB (https://atb.uq.edu.au/index.py?tab=amino_acids), this includes entries
described herein, N- and C- terminal versions of all genetically encoded amino acids and all ncAAs contained within the
SwissSidechain DB (www.swisssidechain.ch).

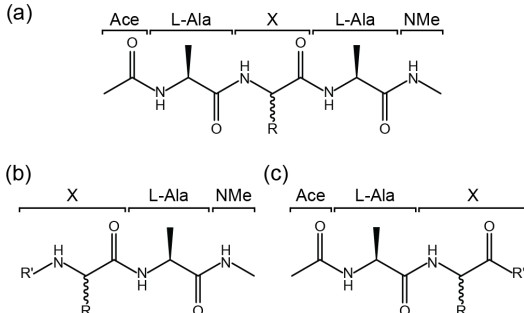

**Figure 2.** Template formats for submission of ncAAs to the ATB. The required templates for a) non-terminal residue, b) N-terminal residue, and c) C-terminal residue in a peptide chain are shown. In each case R represents the sidechain of the ncAA labelled 'X', and in the case of b) and c) R' represents possible modifications of the termini.



### 2.3 Renaming algorithm

When working with ncAAs, it is important to adhere to a consistent set of rules when naming the atoms on the sidechain. Additionally, it is important for the workflow that atom names are unique. We therefore developed a tool to ensure both requirements are fulfilled. In 1969, a set of rules were recommended by IUPAC-IUB for the representation of proteins and peptides (Iupac, 1970). A section of these rules pertains to the labelling of the constituent atoms in amino acids. These rules are adhered to in the presentation of NMR structures of proteins and peptides (Markley et al., 1998).

Non-hydrogen atoms present in the sidechain of amino acids are identified based on the lowest number of bonds that separate them from the $C\alpha$ atom (determined from the connectivity information in the CYANA template or Mol2 format). The order of atoms is indicated using the Greek alphabet (using corresponding Roman characters in files). The first atom connected to the $C^\alpha$ is $C^\beta$, $C^\gamma$ is two bonds away and so on. While seemingly straightforward, complexities of this naming system arise for branched amino acid sidechains, which are common in many ncAAs. In the event of a branch, where two atoms are the same number of bonds away from the backbone and therefore are assigned the same Greek letter, chain priority is assigned based on the Cahn–Ingold–Prelog priority (CIP) rules (Cahn and Ingold, 1951).

An algorithm was written to automatically name the sidechain of amino acids (github.com/ATB-UQ/fixnom). This involves the following steps:

1. The library file containing atom coordinate and connectivity information is read as input (CYANA template format or Mol2 format). The input is parsed and a reduced matrix is formed containing only the heavy atoms and their connectivities (to heavy atoms).
2. The matrix is expanded to introduce dummy atoms to represent unsaturated bonds– these are required for the implementation of the CIP rules. The $C^\alpha$ atom is identified and the matrix reordered based on this information.
3. A connectivity matrix is created describing the distance (in bonds) between any two atoms. The atoms are ordered based on their distance to the $C^\alpha$ atom.
4. At this point the chains are initiated by considering all atoms connected to the $C^\alpha$ position. All (heavy) atoms connected to $C^\alpha$ are given a position and a chain identifier. The position is simply the number of bonds away from $C^\alpha$. The chain identifier follows the CIP rules. The chain with highest priority is assigned the lowest number and additional chains are provided an incremented chain number based on priority. The priority of the chain is determined as follows:
   a) The priority of that atom (atomic number)
   b) The priority of each attached atom one bond away. If all connected atoms have the same priority, then evaluate all atoms two bonds away and so on.
   c) If the priority cannot be resolved by the above method, the atomic coordinates of the atoms are considered and the R/S position of the branching atoms are used to assign priority. This last step is only required for cases of tetrahedral atoms containing two identical chains (ignoring the preceding atom in the chain). If there are three identical chains their identity is arbitrary by rotation (and no further action is taken). Practically this is implemented by using the method of Cieplak and Wisniewski (Cieplak and L., 2001), where the determinant of the $4 \times 4$ matrix formed by the atomic coordinates of the atoms (X, Y, Z, 1) in the stereocentre is used to evaluate if an atom is clockwise or counterclockwise in position with respect to a geminal atom. The atom with the highest priority is placed as the bottom row of the matrix, the two top rows are then occupied by the two atoms being evaluated. As noted by Cieplak and Wisniewski the position or the identity of the third row does not affect the handedness of the atoms being evaluated, and this atom can be "placed" at the position of the central atom. In



their implementation they do this for cases where the fourth atom is a hydrogen (often removed in databases) and
show that this is valid since this atom will always have a lower priority than the other atoms. Similarly, in our
case, we do not need to know what this atom is since it will always have a lower priority than the atom preceding
the branch point (according to the CIP elaboration by Markley et al. (Markley et al., 1998), the atom closest to
the Cα atom always has the highest priority), thus we can simply fill the third row with the atomic coordinates
of the central atom.
5.   The above procedure assigns chain numbers to atoms that are one bond ahead of the atom being considered in an

214       incremental (one pass) approach. This requires that at each increment, the atoms that have already been assigned

215       priority (ahead of the atom being evaluated) are reordered based on their assigned priority in the previous step.

6.   A situation can arise that an atom is evaluated twice if it joins two branches, as is the case for rings. In these cases,

217       the atom is given the lower chain ID between that already assigned to it, and what would be assigned to it had it not

218       been a joining atom.

Once all the chains have been generated, a procedure for re-evaluating identical chains (due to symmetry) in different parts of
the molecule is applied. Here the chain number is reordered based on the priority of the chain from which they branch. This is
not explicitly defined by the CIP rules but is required to ensure that if a branch contains two identical aromatic rings, the chain
numbers in each aromatic ring are consecutive (i.e., to avoid a ring having chain identifiers 1 and 3 or 2 and 4).
Once all chains have been created, the chain number and the distance from the $C^\alpha$ atom are used to generate the correct atom
names. Note, if a position exists that belongs to chain 1 and only one (heavy) atom occupies this position (number of bonds
away from $C^\alpha$), this atom is not given a chain number, and only the Greek alphabet character corresponding to the bond position
is used. Finally, hydrogen (and pseudo) atoms are added. For cases where two hydrogens are attached to a tetrahedral carbon,
the stereochemistry of these is used to assign priority (using the same method as outlined above in step 4c).
This algorithm has been implemented in the ATB using an open-source standalone Perl script developed in-house
(github.com/ATB-UQ/fixnom) and is used to apply IUPAC atom naming to all amino acid building block outputs. This
algorithm is used internally in the ATB to rename atoms in all output files, including PDB files that can be used to define
atomic templates in NMR analysis software such as CCPN (available in v2 and planned for v3 (Skinner et al., 2016; Vranken
et al., 2005)). An example of a complex ncAA named by the algorithm is shown in Fig. 3.






**Figure 3.** Generation of template files following the IUPAC atom naming conventions for amino acids. As an example, the
atom names of the ncAA D-phenyl-Gly* (ATB MOLID 606467), present in vancomycin, were automatically generated using
an in-house algorithm. Greek letters are used to signify the number of bonds between a given atom and the $C^\alpha$ atom, with β
denoting one bond, γ denoting two bonds, continuing to o denoting 14 bonds away. Chain priority is shown by the number
next to the Greek letter. In this diagram several important priority assignment examples can be seen. For example, at the branch
of $C^\theta$ to $C^\iota$ and $O^\iota$, priority is assigned to the chain with the heavier atom, oxygen, hence the priorities reflect $C^{\iota 3}$ and $O^{\iota 1}$.

240

**2.4 ATB – CYLIB – CYANA pipeline**

241

Due to the torsion angle dynamics algorithm used by CYANA to efficiently sample configurations, the parameters used to
describe amino acids are arranged according to a tree structure with the N-terminus at the base and the sidechains (and C-
terminus) as terminating branches (Guntert and Buchner, 2015). Because of the inherent complexities in representing
molecules in this manner, the ATB utilises the CYLIB application to produce a CYANA library file for amino acid building
blocks (Yilmaz and Guntert, 2015). This is achieved by excising the target group (X) from the templates outlined in Fig. 2 and
producing a CIF file containing the variables, amino acid backbone atoms (and atom names) to match amino acid structures
within the CCD (Westbrook et al., 2015). The resulting CCD compatible amino acid CIF file is passed to CYLIB with the
arguments required for sidechain, C- or N-terminus groups and the resulting files made available as downloadable files on the
ATB site. Note that in cases where sidechains or termini contain cyclic elements (within an amino acid), CYLIB also produces
a restraint macro to close the cycle which is also provided as a downloadable file.




**2.5 CYANA Lib Linker distance restraints**
To enable amino acid sidechain and backbone cyclisation within CYANA, the ATB provides a tool to generate distance
restraints, linking statements and modified library files called CYANA Lib Linker (atb.uq.edu.au/cyana_linker).
The CYANA Lib Linker takes two input CYANA library files both of which contain the two atoms that will form the link
between the two amino acids plus at least a one atom extension beyond the linking bond (e.g. a disulfide bridge could be
formed by linking two sidechains of cysteines in the form $C^\alpha$-$C^\beta$-$S^\gamma$-$S^{\gamma'}$-$C^{\beta'}$) – see Fig. 4. To make the process generic for any
linkage, templates are required for each side of the link (in the disulfide bond example the same template is used twice as
input). For each template the following must then be defined: (1) "Residue index", which is the residue number in the intended
peptide sequence; (2) "Linking bond", defined by the two atoms involved in the sidechain linkage ($S^\gamma$-$S^{\gamma'}$); (3) "overlap atoms"
defined as atoms that exist in both templates (these atoms may have different names – in the disulfide bond example, this
would be a process of pairing atoms $C^\beta$ of one template with the $C^{\beta'}$ of the other and $S^\gamma$ of one with the $S^{\gamma'}$ of the other). Once
all of the above is satisfied, the algorithm edits the input template files by altering the atom type of the overlap atoms (and
attached hydrogens) in the library extension ($S^{\gamma'}$-$C^{\beta'}$) as "DUMMY" (excluding PSEUDO atoms which are not altered). An
upper distance is defined for each pair of corresponding overlap atoms with a limit of 0.04 Å (chosen arbitrarily, must be larger
than 0 and within experimental uncertainty) and assigned a weight of 10 (i.e. equivalent to 10 NOEs in CYANA). CYANA
Lib Linker also produces a link statement that removes the repulsion term between the bound atoms (between the two $S^\gamma$ atoms
from each template) within CYANA. The link statement is included in the sequence file input of CYANA.

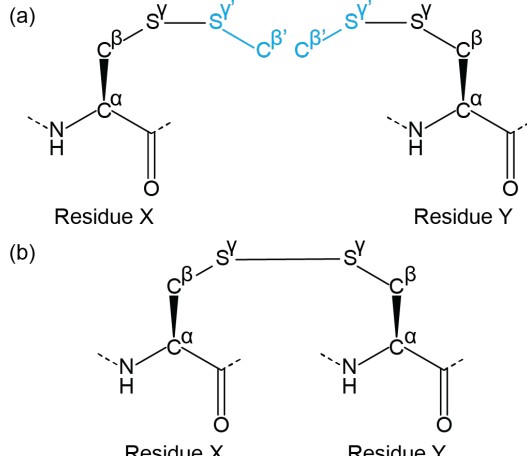


**Figure 4.** Example of a sidechain cyclisation using the Lib Linker for CYANA. Two extended ncAA template files are
generated representing either side of the linkage (extension shown in cyan), including the linking atom and neighbouring atoms
that define the chemical environment of the linked atoms. a) For a disulfide bond, the linked residues are the same on either
side and a single template can be selected as both input files for Lib Linker (Residue X and Y). Each side is, however, defined
as having different positions in the peptide chain. The connecting atoms ($S^\gamma$ from each template) and overlap atoms ($C^\beta$ and $S^\gamma$
overlap with $C^{\beta'}$ and $S^{\gamma'}$ of the other template, shown in black and cyan respectively) are provided as user defined input to Lib
Linker. b) Lib Linker generates two templates (identical in this case), where only the atoms from each side of the linking bond
are included. Restraint files are also generated that can be used directly for CYANA structure calculations.



**2.6 ATB-CNS Pipeline**

MD force fields have been incorporated into several NMR structure determination packages including XPLOR (currently distributed as Xplor-NIH), ARIA and CNS (Schwieters et al., 2003; Schwieters et al., 2006; Bermejo and Schwieters, 2018). For non-backbone-modified ncAAs lacking sidechain cyclisation, implementation into CNS forcefields for structure calculations is straightforward. The ncAA is built and processed by the ATB as described above. To maintain compatibility with the current protein force field used by CNS, the amino acid building block files produced by the ATB retain the standard CNS atom types as well as the bonded and non-bonded parameters for atoms involving protein backbone definitions (N, HN, CA, HA, CB, O). This allows standard linkage statements to be used for producing the molecular template. The new topology and parameters can simply be added to the standard files without modification. New atom types are defined for all other atoms in the ncAA allowing the new geometry and charges to be defined. In order to conserve the net charge when combining the CNS protein backbone charges with the ATB charge model, any residual charge is simply added to the non-backbone atom with the largest magnitude charge. Note that while the duplication of atom types is prevented between ATB outputs and the standard CNS protein force field, atom types defined within separate ATB outputs are not unique. In cases where multiple and distinct ATB parameterised groups are being used within a single peptide, manual resolution of atom types to ensure they are unique may be required.

For sidechain-linked ncAAs, a manual addition of the cross link is required. As for the CYANA approach, generating building blocks with overlapping atoms allows all aspects of the required geometry to be defined and topology and parameters directly added to the existing forcefield. A specific linking statement that removes the extra atoms, modifies charges if required, and adds the required bonds, angles and improper dihedrals (analogous to how disulfide bonds are implemented), can subsequently be written by the user. Similarly, ncAAs that include backbone modifications, and thus cannot be modelled using the existing standard linkage statements for creating peptide bonds, also require manual modification of the peptide bond linkage statement to be incorporated and modelled correctly.

**3 Results**

**3.1 Disulfide bonds**

The handling of sidechain linkages described here is a departure from the default approach used in CYANA for linking disulfide bonds. Currently, disulfide bonds are defined using a special template file called CYSS, where the template contains all atoms up to the linking sulfur atom. A set of upper and lower, one- and two- bond distance limits are then imposed to maintain an appropriate geometry around the introduced bond. One problem with this approach is that the $\chi_2$ and $\chi_3$ torsion angles of disulfide bonds are not defined in the template and are therefore neither explicitly subject to the CYANA torsion angle search, nor can they be easily constrained to specific values.

To validate the use of the new template (CYSX), we recalculated the structures of a number of disulfide-rich peptides resolved in our group (e.g. 2KSL, 5LIC) using the new template (data not shown). In general, the recalculated structures are very similar to those previously calculated using the CYSS template and distance restraints to define the disulfide bond. While it is beyond the scope of this work to investigate in detail how the subtle differences between the two methods affect the quality of the calculated structures, we can make some general observation about the process. We note that using the CYSX template instead of the CYSS template in CYANA requires only trivial changes to the associated files. Specifically: (1) CYSS is replaced with



CYSX in the sequence file (the "link" statement which removes the repulsion between the connected sulfur atoms is used in
both approaches and requires no further changes); (2) CYSX is appended to the existing CYANA library; (3) the old restraints
for upper and lower distance limits to define the disulfide bond are removed and replaced with the new short upper distance
limits between "overlap atoms"; (4) after structure calculation the dummy atoms are removed and the CYSX name changed
to CYS. The last point can be achieved by adding four lines of commands to the end of the CYANA structure calculation script
(using existing CYANA functions). Warnings may occur due to a conflict between CYSX and CYSS in other restraints files
(i.e. dihedral angle restraints), but these can be ignored. The required template file for CYSX, the associated distance restraints
and CYANA commands for removing the DUMMY atoms and producing a PDB file suitable for subsequent analysis and
deposition to the PDB are provided in the supplementary section and our GitHub repository (github.com/ATB-UQ).

### 3.2 ncAAs in complex natural products

To demonstrate the capability and workflow, we have selected three examples of ncAA-containing peptides and generated the
additional library files required for structure calculation by CYANA. These peptides were selected to demonstrate particular
aspects of our pipeline and highlight potential practical applications. For each example, ncAA templates and restraints were
generated following the above procedure, and we demonstrate that these lead to chemically sound structures during
unrestrained structure calculation in CYANA.
Tyrocidine is a cyclic decapeptide antibiotic (Loll et al., 2014) (Fig. 5a) and contains one ncAA which is currently not present
in the standard CYANA library. The missing ornithine (at position 9) library file was generated by building the tripeptide:
Ace-Ala-Orn-Ala-NMe, in PyMOL (using the PyMOL Builder) and the resulting PDB file was saved and submitted to the
ATB, (MOLID 467880). The resulting ATB entry provides all necessary files for NMR structure calculation. The CYANA
template generated by the ATB was appended to the CYANA library. Existing procedures in CYANA were used to create a
backbone linkage to cyclise the peptide chain. Unrestrained CYANA calculations were performed, resulting in an ensemble
of chemically feasible structures (without NMR restraints). An example structure is shown in Fig. 5b.
Cyclosporine (Corbett et al., 2021) is an 11-residue backbone-cyclised peptide commonly used as an immunosuppressant to
treat rheumatoid arthritis and Crohn's disease, and in the prevention of organ rejection in transplants. Cyclosporin contains six
ncAAs (Fig. 5c), five of which are not present in the standard CYANA library. The ncAAs are: D-alanine (residue 1), (4R)-4-
[(E)-2-butenyl]-4,N-dimethyl-L-threonine (MeBmt) (residue 5),α-aminobutyric acid (Abu) (residue 6) as well as three N-
methylated amino acids (sarcosine, residue 7; N-methyl-valine, residue 4; N-methyl leucine, residues 2, 3, 8, 10). Overall,
cyclosporine required five new library files to be defined and appended to the CYANA library (D-alanine can be generated
from alanine using the "library mirror" command in CYANA). A tripeptide template was generated for each and submitted to
the ATB. The ncAAs needed to describe cyclosporine are now available with the following MOLIDs: N-methyl leucine,
1175924; N-methyl-valine, 1175930; MeBmt, 1175933; Abu, 1175938; and sarcosine, 1175941. Feasible structures were
again obtained following unrestrained torsion angle dynamics simulations using CYANA (Fig. 5d).
The final example, vancomycin, is a glycopeptide antibiotic that has been included on the WHO list of most essential medicines
(Schafer et al., 1996). It was, until recently, used as an antibiotic of last resort. This tricyclic heptapeptide is an example of a
non-ribosomal peptide. Five of the seven amino acids in vancomycin are involved in sidechain-sidechain branches. These
residues must be generated individually and subsequently processed through the CYANA Lib Linker algorithm to define

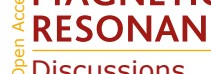

overlap atoms and distance restraint files. Additional distance restraint files are also required to close the sugar rings present
in residue four.
Three sidechain links are present in vancomycin, the first (link 1) between residues 2 and 4, the second (link 2) between
residues 4 and 6 and the third (link 3) between residues 5 and 7. Templates were generated with extensions beyond the linking
atom (including the aromatic rings on either side of the ether bond – MOLIDs: 1212202 (residue 2) and 1212203 (residue 4)
for link 1. The overlap atoms included the bonding atom ($O^{\eta 3}$ and $C^{\delta 1}$ in link 1 and $C^{\delta 2}$ and $O^{\eta 3}$ for link 2), as well as all atoms
one bond away from the linking atom ($C^{\zeta 3}$, $C^{\iota 3}$, and $C^{\iota 4}$ in residue 2; and $C^{\zeta 1}$, $C^{\varepsilon 3}$ and $C^{\gamma 1}$ in residue 4). For link 2, one additional
extended library file (MOLID: 1213034) was generated for residue 6, and the following overlap atom mapping was used: $C^{\gamma 2}$,
$C^{\varepsilon 3}$ and $C^{\zeta 2}$ in residue 4; and $C^{\iota 4}$, $C^{\iota 3}$ and $C^{\zeta 3}$ in residue 6. Note that the modified residue 4 from link 1 is used as input when
creating link 2. For link 3, between residues 5 and 7, two additional library files were generated, MOLIDs 1212698 and
1212205 respectively. Link 3 exists between the bonding atoms $C^{\delta 1}$ and $C^{\gamma 3}$ with overlap atom mapping: $C^{\gamma 1}$, $C^{\varepsilon 1}$, $C^{\zeta 3}$ and $C^{\zeta 4}$
in residue 5; and $C^{\varepsilon 5}$, $C^{\varepsilon 3}$, $C^{\delta 1}$ and $C^{\beta}$ in residue 7. Finally, a template was produced for the N-terminal residue (MOLID
1212206). Using these template files, unrestricted torsion angle dynamics were performed in CYANA (Fig. 5f).
Vancomycin contains two sugar rings, and in CYANA each must be "closed" using an internal ring-closure process. CYLIB
automatically generates the required CYANA commands for closing rings, however, it currently cannot handle multiple ring-
closures, such as those for residue 4 of vancomycin. This required an additional ring-closure statement to be added manually.
An additional manual step involved adding a torsion angle that defines the angle between the two aromatic rings that are
directly fused (between residues 5 and 7).



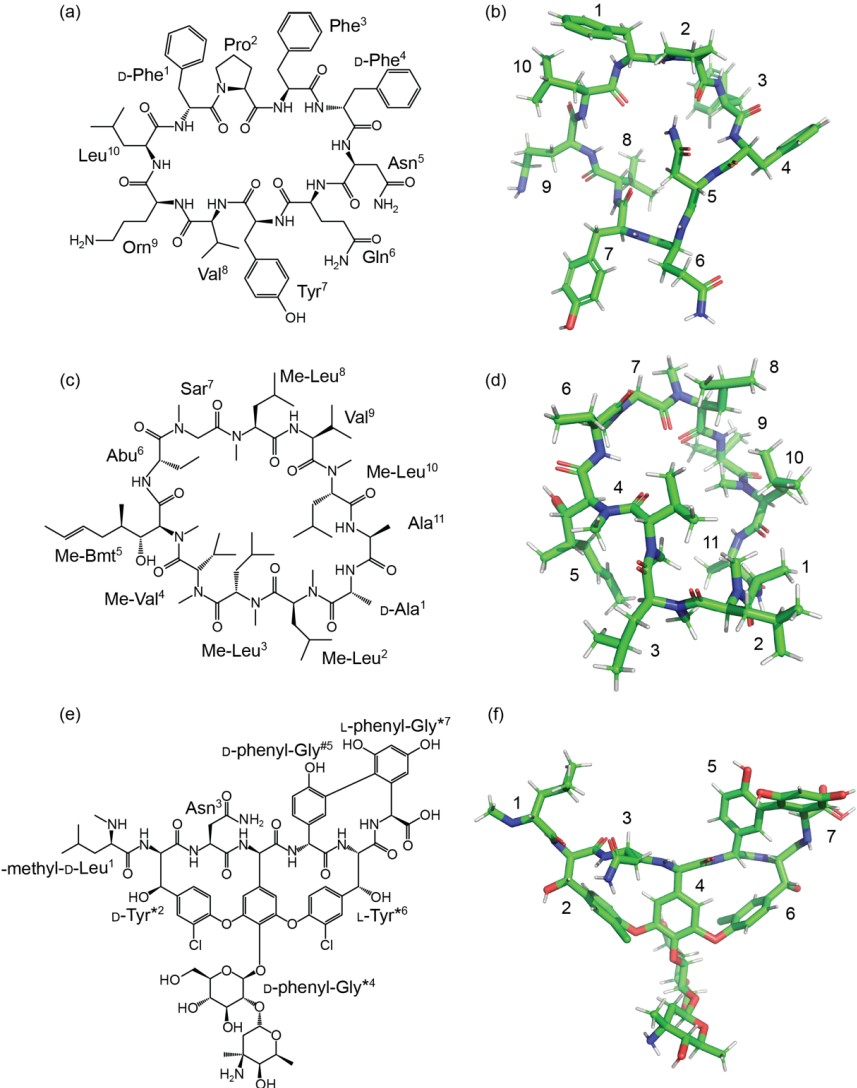

**Figure 5.** Structures of natural products generated from un-restrained CYANA calculations using ncAA templates generated by the ATB as described herein. (a) 2D and (b) 3D structure of tyrocidine containing a single ncAA (ornithine). (c) 2D and (d) 3D structure of cyclosporine. The sequence of cyclosporine contains five ncAA, N-methyl-leucine (Me-Leu), N-methyl-valine (Me-Val), 4R)-4-[(E)-2-butenyl]-4-methyl-L-threonine (Me-Bmt), α-Aminobutyric acid (Abu), and sarcosine (Sar). (e) 2D and (f) 3D structure of vancomycin. The amino acids of vancomycin have been abbreviated in some cases. D-Tyr*2 is m-chloro-β-hydroxy-D-Tyr. D-phenyl-Gly*4 is (2-[α-4-L-epi-vancosaminyl]-β-1-D-glucosyl)-D-phenyl-Gly. D-phenyl-Gly#5 is p-hydroxy-D-phenyl-Gly. L-Tyr*6 is m-chloro-β-hydroxy-L-Tyr and L-phenyl-Gly*7 is m,m,-dihyroxy-L-phenyl-Gly. The 3D structures were generated by CYANA based on the templates obtained by the ATB – CYLIB – CYANA pipeline without experimental restraints.



### 3.3 Practical application using an engineered peptide

Advances in peptide chemistry are rapidly expanding the chemical space accessible to high-throughput peptide synthesis methods. Recently, a systematic approach was taken to explore sidechain stabilisation of a segment of the amyloid precursor protein (APP; [1]NGYENP**T**YKF**F**E[12]) into the conformation it adopts when bound to the phosphotyrosine binding (PTB) domain of Mint2 (Bartling et al., 2022). The method described herein was developed to solve the structure of four peptides with different sidechain linkages (Bartling et al. under review). One of these peptides is sidechain-cyclised through ring closing metathesis (RCM) between residues 7 and 11. The 3D structure of this peptide has been reported elsewhere (Bartling et al. under review), and was solved using the methods described herein. The sidechain cyclisation following RCM was, however, treated using a similar approach to that currently used in CYANA for fusing sulfur atoms in disulfide bonds (i.e. using truncated template files). Here, we revisit this structure and perform the sidechain fusion using the new approach described here. We further demonstrate how CNS templates, generate by the ATB, can be used for water refinement of the reported CYANA structure.

First, Thr7 and Phe11 were replaced with the α-methyl-substituted olefin-bearing ncAA named pentenyl alanine "PAL" (MOLID 929126). This is an extended form of the PAL sidechain as shown in Fig. 6. The CYANA Lib Linker was then used to modify the template (DUMMY and overlap atom definition) and to generate appropriate upper distance restraints. For the terminal residues, templates corresponding to the N-terminal Asn and C-terminal Glu (MOLIDs 1162954 and 1159438) with their respective amino and carboxy acid termini were also generated.

To assign the magnetic resonances of the peptide, CcpNmr Analysis 2.4.1 was employed (Vranken et al., 2005). The atomic composition of individual ncAAs in CcpNmr were defined by uploading the ATB-generated coordinate files into its molecule library. Resonance assignments were obtained using a combination of 2D $^1$H-$^1$H TOCSY, 2D $^1$H-$^1$H NOESY and natural abundance 2D $^1$H-$^{15}$N and $^1$H-$^{13}$C HSQC. The $^1$H chemical shifts were calibrated with the reference to the water chemical shift while $^{13}$C and $^{15}$N chemical shifts were calibrated indirectly with the reference to $^1$H. Cross-peaks from the 2D $^1$H-$^1$H NOESY (mixing time of 350 ms) were manually picked to generate a list of interproton distance restraints. TALOS-N was used to derive dihedral angle restraints. As TALOS-N does not recognize ncAAs, we replaced the ATB generated residue codes for the terminal residues with N and E respectively, in the TALOS-N input file. The angle restraint range was set to twice the estimated standard deviation.

To perform calculations using CYANA the following files were prepared using the ATB, CcpNmr and TALOS-N: (i) a sequence file listing the amino acid sequence of the peptide, (ii) a chemical shift file listing the chemical shifts of all assigned atoms, (iii) a peak list of the 2D $^1$H-$^1$H NOESY spectrum containing the chemical shifts and calibrated peak intensity (height or volume) of each peak, (iv) an angle restraint file derived from TALOS-N, (v) a CYANA library file specific for the ncAA templates acquired from the ATB, and (vi) restraint files specific for the sidechain linkage generated by CYANA Lib Linker. The sequence file and peak list (distance restraints) were directly exported from CcpNmr in the CYANA-compatible XEASY format. The chemical shifts of protons in the peptide, were first exported from CcpNmr in BMRB format (any other formats would omit the entries for any ncAAs – a current CcpNmr limitation). The shift list in BMRB format was imported into CYANA and pseudoatoms were added using an internal CYANA command. CYANA (v. 3.98.13) was then used to automatically assign the peak list, extract distance restraints, and calculate 200 structures from which 20 structures with lowest target function values were selected to represent the structure ensemble of the peptide.

The output of the CYANA structure calculation was used to perform water-refinement in CNS. The CNS topology and parameter output for PAL was added to the standard forcefield and this residue included at both positions to be linked. A



MAGNETIC RESONANCE
Discussions
linkage statement deleting the extra atoms was subsequently introduced to generate a complete molecular template file with
all the atoms and restraints required to define the geometry of the residue. Because of the additional backbone methyl group,
a custom peptide bond linkage statement was also constructed and used to create the residue links on either side of each ncAA
(both custom linkage statements are also available on github.com/ATB-UQ). Structures were calculated and minimised in
water using the experimental distance and angle restraints, resulting in a well-defined family of structures with excellent
geometry and no violations of the experimental data (Fig. 6c).

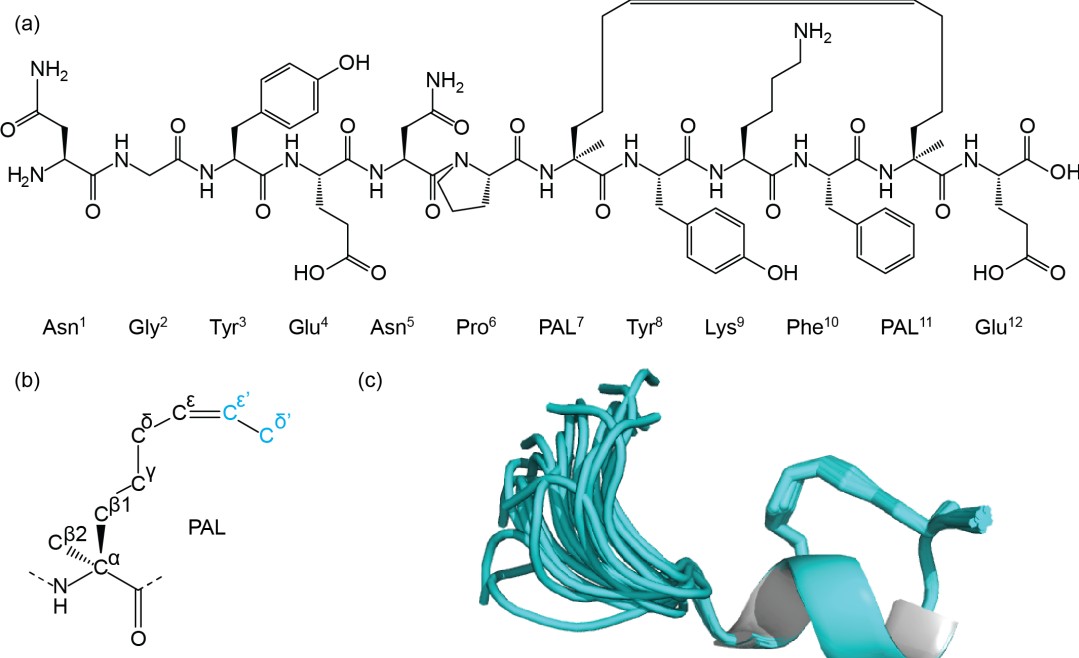


**Figure 6.** Figure showing the chemical structure of a synthetic peptide (a) that mimics the MINT-2 bound form of the amyloid
precursor protein (APP). The helical structure in the C-terminal tail of the peptide is stabilised through sidechain cyclisation
using ring closing metathesis (RCM). The RCM linkage was modelled using an extended pentenyl alanine (PAL) library entry
(b) generated by the ATB. The ATB generated library was further processed by the CYANA Lib Linker interface of the ATB
to generate a new template where the two overlap atoms $C^{\varepsilon'}$ and $C^{\delta'}$ are converted to DUMMY atoms and used to constrain
the sidechain around the double bond. The 3D structure of the peptide was determined using CYANA and then further refined
in explicit water using CNS (c) with topology and parameter files generated by the ATB. The structure shows stabilisation of
the helical motif (cartoon) and the PAL sidechain is shown as sticks.

**4 Discussion**
A workflow has been developed that facilitates the structure determination and refinement of peptides and proteins that contain
ncAAs using popular NMR software, including CYANA and CNS. Methodology has been incorporated into the ATB server,
which allows arbitrary ncAAs to be built as required. New ncAAs are stored on the site and added to a database of entries
containing all of the necessary files for structure determination by NMR spectroscopy. The workflow introduces a number of



automated procedures to improve access to complex ncAAs without manual intervention. An algorithm has also been
developed that can automatically assign IUPAC atom names to ncAAs. It has already been highlighted that there have been
errors within the PDB when naming for diastereotopic atoms (Bottoms and Xu, 2008). The potential explosion in elaborate
sidechains that is possible when introducing ncAAs requires that care is taken when defining atoms, reporting and comparing
data.
We have also introduced into this workflow a stand-alone and general method for introducing sidechain-to-sidechain bonds in
CYANA. A graphical user interface has been incorporated into the ATB to facilitate this process using user supplied template
files (i.e. the files generated within the server). The new sidechain linking procedures uses overlap "DUMMY" atoms to
establish the missing bond in CYANA. This is a departure from the standard method currently used in CYANA for linking
sidechain atoms of pairs of cysteines. Tests of this procedure showed that the structures produced by both the new and old
methods were very similar. Importantly, the new approach allows for definition of the $\chi_2$ and $\chi_3$ torsion angles in cysteine
bridges, enabling these to be sampled by the torsion angle dynamics in CYANA, and for these angles to be defined where
experimental evidence is available (Armstrong et al., 2018; Ramanujam et al., 2019). In the "traditional" approach of defining
disulfide bonds, these torsion angles are not defined and disulfide bond geometries are instead sampled indirectly by altering
the $\chi_1$ angles and the consequences of this on the imposed distance restraints and repulsion terms associated with the other
sidechain atoms. While the result appears to be similar using the two methods in the absence of $\chi_2$ and $\chi_3$ restraints, the new
method will certainly be favourable where such data are available. More generally, this procedure will allow all sidechain links
to include definition of inter-residue torsion angles in CYANA.
The utility of the workflow was demonstrated by building structural models of several natural products in CYANA. In most
cases only a single ncAA is present in a peptide or protein chain, due to incorporation of the ncAA via recombinant expression
methods, via peptide synthesis or chemical modification of specific AAs in the produced peptide or protein. This was captured
by modelling tyrocidine, a short peptide incorporating a single ncAA. In such cases, the method is very robust and little manual
intervention is required (beyond adding the additional templates to the CYANA library). Cases involving multiple ncAA,
especially those containing backbone N-methylation, are very challenging to prepare manually, however, even cyclosporin
poses no issues in our pipeline beyond the once-off generation of the template files. The most challenging example explored,
the antibiotic vancomycin, is an extreme case involving multiple interlocked ring structures formed by linking ncAAs. CYLIB
currently cannot automatically process groups with multiple rings, such as residue 4 in vancomycin which has two sugar
moieties branching from an aromatic ring.  Thus, some manual steps are required to add the extra ring closure statements.
Similarly, the central residue 4 of vancomycin is sidechain-linked to two different residues. This requires serial iterations of
the CYANA Lib Linker interface to generate the required files – i.e. first templates for residues 2 and 4 are used to generate
modified templates, the modified template for residue 4 is then further modified when submitted as the linking residue to
residue 6. Even this challenging molecule could be processed using the tools developed.
The results described above demonstrated that the ncAA templates are compatible with existing libraries and yield high quality
structures. We also showed how one might use the templates in conjunction with experimental data. The APP-derived peptide
that includes an RCM-cyclised sidechain, was used to highlight some practical considerations associated with the pipeline
described in this work. Some manual processing is still required to use our templates with the popular CcpNmr software. While
we were able to create a working solution for v2 of CcpNmr, incorporation of ncAAs within a peptide chain in v3 is currently
not feasible. Other analysis packages such as POKY do not require templates and may be more suited for use with ncAAs (Lee
et al., 2021). We also noted that additional automation was required to appropriately import the output of CcpNmr into



CYANA. Although we have a working solution, it is likely that future development of CcpNmr will address these problems
that exist when working with ncAAs.
The final test involved the refinement of the APP peptide in water using the CNS software. This currently involves manual
creation of the linkage statements. This is relatively straightforward involving a simple modification of the standard peptide
bond and/or the standard disulfide bond definitions. Importantly, because the geometries are already defined in the building
blocks and all parameters required are included in the ATB output, the statement simply has to define which atoms are linked
and infer other geometrical constraints such as the planarity of the double-bond. Once written, these linkage statements can be
used for any variant of a given ncAA. For example, the peptide backbone link used for the "PAL" residue can be applied to
any residue type in which the $H^{\alpha}$ proton has been replaced with a methyl group.
The workflow presented here was developed to cater to the rapid growth in peptide and protein engineering in recent years,
examples including directed evolution mRNA display methods to generate macrocyclic peptide ligands of target receptors
(Goto and Suga, 2021), modified amino acids for structural studies (Mekkattu Tharayil et al., 2022) and high-throughput
chemical synthesis in drug development (Bartling et al., 2022). These developments have increased the interest in solving
structures of ncAA-containing peptides leading us to develop the described method. The protocols and tools we have developed
are designed to be general and to interface with a range of software. This said, it is likely that not all ncAAs that can be
envisaged will be able to be handled with the existing workflow. Nevertheless, the solution we presented in this work not only
addresses a pressing need in the cases of ncAAs but also provides a general framework that can be used to improve the
description of AAs in structure refinement more broadly.

**5 Conclusions**
Peptides containing ncAAs encompass a large pool of biologically active molecules with many potential industrial, agricultural
and pharmaceutical uses. The significant structural diversity found in these peptides presents significant challenges for NMR
spectroscopists when applying existing structure determination tools. This problem requires the development of a set of tools
that can automatically generate molecular representations that have suitable chemical properties. We have here provided a
solution to this problem in the form of an extension to the Automated Topology Builder, which now can produce template
files compatible with most commonly used NMR structure calculation software. We have also ensured that the ncAA templates
generated adhere to IUPAC naming conventions based on the Cahn-Ingold-Prelog priority rules. This extends the utility of
NMR structure calculation to complex natural products, synthetic peptides, and complex, natural and unnatural, post-
translational modifications.
**Code and Data availability**
The ATB server is available publicly at (atb.uq.edu.au). The code used in this work is available via GitHub as cited in the
manuscript (github.com/ATB-UQ).
**Author contribution**
SK, MS, MM and AEM conceived of the project and designed elements of the pipeline. MS implemented the code in the ATB
with input from AEM. SK validated the implementation with input from MM. MS, TL and MM wrote code used in the ATB
workflow. SK, YKYC, ACC, XJ, KJR and MM produced the models presented and analysed the NMR data. CORB and KS



provided the synthetic APP peptide. PG contributed to CYANA template generation and sidechain linking method. SK wrote
the first draft of the manuscript and all authors contributed to the final version.
**Acknowledgements**
This project was supported by funding from the Australian Research Council (DP DP190101177 and DP220103028 to MM.
DP220100896 to AEM and MS), the National Health and Medical Research Council (NHMRC APP1162597 to MM) and the
University of Queensland (Postgraduate Research Scholarship to SK, Research Stimulus fellowship to YC and Development
Fellowship to MM) and in part by the Austrian Science Fund (FWF) (Project P36101-B) to AC.

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
