# Peer review of "Facilitating the structural characterisation of non-canonical amino acids"

_Magnetic Resonance, 2022_

## Author Response (AR1)

Response to reviewers.

We thank the reviewers for their critical review of our work. Reviewer 1 did not provide any specific points to address. Below, we have addressed the points raised by Reviewer 2 and 3.

Reviewer 2

1) While using successive letters from the Greek alphabet to indicate the (smallest) number of bonds separating a sidechain atom from the protein mainchain in an ncAA is clearly consistent with the spirit of the IUPAC recommendations, it seems to me that formally speaking this is a step beyond the IUPAC recommendations themselves that, as far as I am aware, refer only to atoms in the 20 canonical amino acids. I don't believe the two references for the IUPAC recommendations given in the manuscript (Huang et al., 1970 and Markley et al., 1998) describe this additional step to include atom naming in ncAAs; are there other references that could be cited that would formally set such a precedent?

The reviewer is correct, IUPAC have not made a specific recommendation for such cases. We invited IUPAC to review our paper and provide comments on our approach. They responded by confirming that no convention exists and suggested additional references to IUPAC documents, which have been included as follows:

*"line 170 it is IUPAC-IUBMB and you should be using the 1983 document.*
*See https://iupac.qmul.ac.uk/AminoAcid*
*line 559 this reference has been superseded by several later documents. For the latest version see P-9 in the 2013 Blue Book. see https://iupac.qmul.ac.uk/BlueBook*
*"*

2) The Greek alphabet contains only 24 letters, and while this is doubtless sufficient for uniquely naming atoms in the great majority of ncAAs or PTMs following the approach outlined in this paper, it is not sufficient in all cases. For instance, glycans containing linear chains of more than 4 sugars would exceed a 24-atom chain-length limit, and the structure of glycosylphosphatidylinositol (GPI) anchors, which are a not uncommon form of N-terminal PTM, involve more than 40 bonding steps from the polypeptide backbone. Do the IUPAC recommendations have anything to say about atom naming in such cases? Assuming they do not, what do the present authors propose in such cases when the Greek alphabet runs out of letters?

We have expanded the standard nomenclature from 24 to (600) by using 2 letter symbols from position 25 (AA, AB, … AW | BA, BB, … BW |…| WA, WB, .. WW). An explanatory sentence has been added to the manuscript.

"(from position 25 the Greek alphabet is repeated as 2-letter codes i.e. $C^{\alpha\alpha}$, $C^{\alpha\beta}$, $C^{\alpha\gamma}$ until $C^{\omega\omega}$ – expanding the available atoms to 600 in this method)"

We have highlighted the problem to IUPAC. Note our code on GitHub is intended to be dynamic and will be amended to match any official recommendation by IUPAC.

3)  In some of the more detailed sections of the manuscript it becomes apparent that some steps in creating files to represent complicated structures do require some manual intervention, e.g. to complete the bonding scheme for some ring structures.  I can see that it may well be very much more difficult to write software that correctly handles such complications fully automatically, and I don't believe that the need for manual intervention to complete the implementation in such cases is necessarily much of a problem, but I do feel the issue should be more clearly discussed in the manuscript.  The authors could comment briefly on whether they are planning to attempt the automatic handling of such remaining cases, or whether that would be impractical.  I think it would also be helpful to add a short but clear statement in a rather more visible location in the paper as to what are the fundamental limitations on fully automatic operation in the present implementation.

A paragraph has been added to  the discussion regarding both the existing limitations and future directions.

"The manual modifications that were noted in specific cases are largely due to limitations and compatibility issues of the tools used in the pipeline. The treatment of ring closures by CYLIB has been noted previously. The need for a torsion angle between connected aromatic rings is a consequence of existing rules within CYLIB. The limitations within the import function in CcpNmr is subject of current development by the authors of that software. The manual steps required for compatibility with CNS are largely due to the use of atom types to define both Lennard-Jones parameters as well as the bonded terms. Name clashes in the atom type definitions that arise from combining multiple ATB generated building blocks within a single system must be addressed to ensure the intended parameters are used. Further refinement of the ATB outputs to improve compatibility with different packages (e.g., NIH-Xplore, AMBER, ARIA) will be the subject of future work."

4)  It is probably inevitable that the implementation of a new approach such as this is rooted in the environment of the particular program using which it was developed, in this case CYANA.  However, the transfer of the approach to a different program environment is important if the method is to be widely adopted, as presumably the authors of this contribution hope it will be.  It may not be practical to go very far down this road, but it might have been nice to see the method worked through using, for instance, XPLOR-NIH, ARIA or AMBER.  Are there steps for which the use of CYANA or associated programs is currently unavoidable?

In section 2.6 we outline how we use CNS to perform water refinement. The topology and parameter files required by CNS are in principle compatible with XPLOR-NIH, ARIA and AMBER (i.e. these can be used for cartesian based structure calculations without the need to use CYANA). The additional paragraph in the discussion now also emphasises this.

Reviewer 3

1) Figure 2 shows the template for the description of ncAAs. What happens if the nitrogen atom of the peptide bond is not bound to an hydrogen but to a carbon such as in methylated AA or di-amino butyric acid found in some bacterial siderophores or a modified proline?

The template recognition matches the ends of the chain and works inwards, so the amino acid itself only has to start with a nitrogen and end with a carbonyl carbon. In principle, it will also

recognise beta amino acids in this way. We have tested the procedure with amino acids such as proline and its derivatives and it works as intended (see also the link to the amino acid tab of the ATB below).

The final paragraph of the introduction now has a link to the ATB amino acid tab.

2) The statement in the first result paragraph is rather odd:

" In general the recalculated structures are very similar to those previously calculated ...."

I disagree with the statement that it is beyond the scope of the work to compare in detail the results of both procedure. The demonstration that the automated approach delivers the same results as the manual one should be provided in a quantitative way and the origin of possible "subtle" differences should be carefully analysed and addressed. The results of structure calculations should comply with accepted standards showing the tables with structural statistics.

The statement refers to the observation that the differences are within the error/precision of the structure calculation by CYANA. The below table demonstrates this:

Table. Torsion angles of disulfide bonds in venom peptide (Ta1a, PDBID: 2KSL) using either the CYSS template (traditional method) or CYSX template (new) method of joining the sidechain of amino acids in CYANA. The average value is that calculated over 20 structures in the ensemble. All details of the structure calculation are identical to those presented previously (2KSL).[1]

| | CYSS Average | (+/-) | CYSX Average | (+/-) |
|---|---|---|---|---|
| DSB-1 | | | | |
| $\chi_2$ [7] | 72 | 113 | 87 | 88 |
| $\chi_3$ [7-37] | 86 | 120 | 77 | 82 |
| $\chi_2$ [37] | -45 | 84 | -16 | 97 |
| | | | | |
| DSB-2 | | | | |
| $\chi_2$ [23] | -52 | 137 | 9 | 154 |
| $\chi_3$ [23-33] | -8 | 45 | 13 | 53 |
| $\chi_2$ [33] | -40 | 151 | 17 | 151 |
| | | | | |
| DSB-3 | | | | |
| $\chi_2$ [26] | -101 | 113 | -83 | 109 |
| $\chi_3$ [26-46] | -47 | 48 | -38 | 57 |
| $\chi_2$ [46] | -65 | 97 | -46 | 100 |

1.      Undheim EA, *et al*. Weaponization of a Hormone: Convergent Recruitment of Hyperglycemic Hormone into the Venom of Arthropod Predators. *Structure* **23**, 1283-1292 (2015).

The differences in torsion angles obtained using either of the two methods were within the spread of the other. We have edited the text in our revision to say that the structures are comparable and removed references to "subtle" differences. We have rewritten the following question:

"The recalculated structures show no clear differences to those previously calculated using the CYSS template and distance restraints to define the disulfide bond (average $\chi_2$ and $\chi_3$ torsion angles in disulfide bonds, obtained using either method, are within the spread of the equivalent angles generated for each method)."

3) As already mentioned, comparative structural calculations should be provided also for CNS or XPLOR-NIH. It would be very helpful to have the example of a topology entry for a modified amino-acid in one of the routinely used force field of CNS.

This already exists on the amino acids tab of the ATB (https://atb.uq.edu.au/index.py?tab=amino_acids), we also included the files used for the calculation of the stapled peptide on github: https://github.com/ATB-UQ/APP-RCM-CNS-Files. We have include the above links in the revised manuscript and added Table 1.

4) In section 3.3, the authors present a practical application on a stapled peptide. Details that are provided should be displaced in the method section rather. As for other examples, a table recapitulating structural statistics should be provided. It would also be interesting to detail how the cis-trans isomery of the double bond is defined from the input structure.

This an oversight on our behalf, and the below table will be added to the results section (based on CNS calculation in water) in the revised manuscript.

| Table: Structural statistics from CNS calculations | |
| --- | --- |
| **Energies (kcal mol$^{-1}$)** | |
| Overall | -375.20 ± 14.91 |
| Bonds | 11.05 ± 1.48 |
| Angles | 45.68 ± 4.92 |
| Improper | 8.00 ± 1.45 |
| Dihedral | 35.48 ± 1.03 |
| van der Waals | -64.21 ± 5.85 |
| Electrostatic | -411.82 ± 22.23 |
| NOE | 0.22 ± 0.03 |
| cDih | 0.40 ± 0.25 |
| **Ramachandran statistics** | |
| Ramachandran favoured (%) | 90.0 ± 12.57 |
| Ramachandran outliers | 0 |
| **Atomic RMSD residues 4-12 (Å)** | |
| Mean global backbone | 0.32 ± 0.13 |
| Mean global heavy | 1.15 ± 0.23 |
| **Experimental restraints** | |
| Distance restraints | |
| Short range (i–j < 2) | 158 |
| Medium range (i–j < 5) | 71 |
| Long range (i–j ≥ 5) | 0 |
| Hydrogen bond restraints | 0 |

| | |
|---|---|
| Total | 229 |
| Dihedral angle restraints | |
| φ | 8 |
| ψ | 7 |
| χ1 | 4 |
| Total | 19 |
| **Violations from experimental restraints** | |
| NOE violations exceeding 0.2 Å | 0 |
| Dihedral violations exceeding 2.0° | 0 |

The trans isomer was determined by analysis of the J-coupling and the NOE patterns. Further details of this are provided in a related manuscript where the initial structural characterisation of this series of compounds is included (Bartling et al. J Med Chem). The manuscript has been accepted pending minor changes and we have updated the reference with the following DOI (https://doi.org/10.1021/acs.jmedchem.2c02017) to be made available shortly.

5) It would be very interesting to provide an example where fluorinated amino-acids are incorporated in a peptide or a protein.

There are many fluorinated building blocks in the amino acid page, for example, pentafluoro-phenylalanine:
https://atb.uq.edu.au/molecule.py?molid=1210306#panel-nmr_refinement

There are already examples of chlorine containing peptides in the manuscript and replacing this with a fluorine would follow the exact same procedure, there is no difference as far as our pipeline is concerned. We encourage users to try the many different functional groups (F, I, Br, NO2, CHO etc.) that we do not cover in the manuscript.